# Function Is More Reliable Than Quantity to Follow Up the Humoral Response to the Receptor-Binding Domain of SARS-CoV-2-Spike Protein after Natural Infection or COVID-19 Vaccination

**DOI:** 10.3390/v13101972

**Published:** 2021-09-30

**Authors:** Carlos A. Sariol, Petraleigh Pantoja, Crisanta Serrano-Collazo, Tiffany Rosa-Arocho, Albersy Armina-Rodríguez, Lorna Cruz, E. Taylor Stone, Teresa Arana, Consuelo Climent, Gerardo Latoni, Dianne Atehortua, Christina Pabon-Carrero, Amelia K. Pinto, James D. Brien, Ana M. Espino

**Affiliations:** 1Department of Microbiology and Medical Zoology, University of Puerto Rico-Medical Sciences Campus, San Juan, PR 00936, USA; lorna.cruz@upr.edu (L.C.); teresa.arana@upr.edu (T.A.); 2Unit of Comparative Medicine, University of Puerto Rico-Medical Sciences Campus, San Juan, PR 00936, USA; petraleigh.pantoja@upr.edu (P.P.); crisanta.serrano@upr.edu (C.S.-C.); tiffany.rosa@upr.edu (T.R.-A.); albersy.armina@upr.edu (A.A.-R.); 3Department of Internal Medicine, University of Puerto Rico-Medical Sciences Campus, San Juan, PR 00936, USA; 4Department of Molecular Microbiology and Immunology, Saint Louis University, St. Louis, MO 63104, USA; taylor.t.stone@slu.edu (E.T.S.); amelia.pinto@health.slu.edu (A.K.P.); james.brien@health.slu.edu (J.D.B.); 5Blood Bank Medical Center, Medical Center, San Juan, PR 00936, USA; consuelo.climent@asem.pr.gov; 6Banco de Sangre de Servicios Mutuos, Guaynabo, PR 00968, USA; glatoni@bssmpr.com; 7Puerto Rico Science, Technology and Research Trust, San Juan, PR 00927, USA; dianne.atehortua@prcci.org (D.A.); cpabon@prcci.org (C.P.-C.)

**Keywords:** SARS-CoV-2, COVID-19 vaccine, neutralization, serology, protection

## Abstract

Both the SARS-CoV-2 pandemic and emergence of variants of concern have highlighted the need for functional antibody assays to monitor the humoral response over time. Antibodies directed against the spike (S) protein of SARS-CoV-2 are an important component of the neutralizing antibody response. In this work, we report that in a subset of patients—despite a decline in total S-specific antibodies—neutralizing antibody titers remain at a similar level for an average of 98 days in longitudinal sampling of a cohort of 59 Hispanic/Latino patients exposed to SARS-CoV-2. Our data suggest that 100% of seroconverting patients make detectable neutralizing antibody responses which can be quantified by a surrogate viral neutralization test. Examination of sera from ten out of the 59 subjects which received mRNA-based vaccination revealed that both IgG titers and neutralizing activity of sera were higher after vaccination compared to a cohort of 21 SARS-CoV-2 naïve subjects. One dose was sufficient for the induction of a neutralizing antibody, but two doses were necessary to reach 100% surrogate virus neutralization in subjects irrespective of previous SARS-CoV-2 natural infection status. Like the pattern observed after natural infection, the total anti-S antibodies titers declined after the second vaccine dose; however, neutralizing activity remained relatively constant for more than 80 days after the first vaccine dose. Furthermore, our data indicates that—compared with mRNA vaccination—natural infection induces a more robust humoral immune response in unexposed subjects. This work is an important contribution to understanding the natural immune response to the novel coronavirus in a population severely impacted by SARS-CoV-2. Furthermore, by comparing the dynamics of the immune response after the natural infection vs. the vaccination, these findings suggest that functional neutralizing antibody tests are more relevant indicators than the presence or absence of binding antibodies.

## 1. Introduction

The COVID-19 pandemic presents an unprecedented challenge to the scientific community. As of today, more than 227 million persons have been exposed to the virus, resulting in more than 4.5 million deaths (https://www.worldometers.info/coronavirus? (accessed on 15 September 2021)). At the same time, it is advancing our collective knowledge in molecular biology, epidemiology, and immunology at an unprecedent accelerated speed. One of the crucial questions still under scrutiny is the magnitude and durability of the immune response to natural infection with SARS-CoV-2, especially given the fact that virus-specific antibody (ab) responses are relatively short-lived following SARS-CoV and common cold coronavirus (CCC) infections (https://doi.org/10.1016/j.immuni.2020.05.002 (accessed on 15 September 2021)). Further complicating this scenario is the recent availability of new vaccine formulations, which are accessible to both previously infected and immunologically naïve individuals. The kinetics of the humoral response in vaccinees, both with and without prior SARS-CoV-2 exposure, is an area of active research with many outstanding questions. Most of the public attention has been focused on the humoral immune response and the discussion is centered in how effective the immune response to natural infection is or to the vaccines depending on the presence or absence and durability of the antibodies. However, the immune response to any infectious agent or vaccine is more complex, and in addition to the antibodies, includes mainly the innate and the cellular-mediated immune responses [1,2,3]. The cellular-mediated immune response, in particular, is proven to be highly effective and durable against the SARS-CoV-2 natural infection, including the variants, or COVID-19 vaccination [4,5,6,7,8].

To begin to address these questions, we followed a cohort of 59 individuals (volunteers or convalescent plasma donors) at different time points following natural infection with SARS-CoV-2. In addition, we chose a set of seven of those individuals plus three additional subjects (*n* = 10) which we then compared with 21 uninfected vaccinated subjects (*n* = 21). Serum samples for both vaccinated groups were collected between 12 and 28 days after each of the two doses of mRNA vaccine and a third sample was collected between 19 and 83 days after the second dose. Because the limited period of SARS-CoV-2 circulation, studies on the quantity, quality, and extent of long-term memory responses are still underway. Recent works on the durability of the humoral immune response after the natural infection with SARS-CoV-2 showed the presence of neutralizing antibodies for several months [4,9,10,11,12] or the persistence of IgG responses over the first few months after infection, which is strongly correlated with neutralizing antibody titer [11,13]. Since the onset of the COVID-19 pandemic, functional neutralization assays using serum antibodies has been severely limited due to the requirement for a biosafety level 3 (BSL-3) facility to grow SARS-CoV-2. However, in a relatively short period of time, several surrogate neutralization assays have become available with an excellent performance profile when compared to the classical focus reduction neutralization test (FRNT) [11,14,15,16,17,18]. For these studies, we chose the cPass SARS-CoV-2 Surrogate Virus Neutralization Test Kit (GenScript, USA) which measures the interaction of purified SARS-CoV-2 S protein receptor-binding domain (RBD) with the extracellular domain of the human ACE2 receptor [18]. Other groups reported that this assay showed the best sensitivity and the lower false negative rate compared to five other assays [17]. Furthermore, this assay was granted an emergency use authorization (EUA) by the Federal Drug Administration (FDA) for the detection of SARS-CoV-2 neutralizing antibodies. Interestingly, we detected a small number of cases (*n* = 6) where neutralization activity was still present, although S-specific IgG titers were undetectable by our method (OD < 0.312).

Recently, debate has centered around the efficacy of the natural immune response to SARS-CoV-2 vs. mRNA vaccines. Our work—which examines patients in a predominantly Latino population—confirms that following a natural infection neutralizing antibody titers remained detectable at high levels for four to seven months. We also demonstrate that the quantity and the quality of the antibody response induced by the natural infection is significantly higher in titer of both binding and neutralizing antibodies when compared to the response induced by mRNA vaccination. There is limited information regarding the magnitude of the immune response to vaccines against SARS-CoV-2 in naïve vs. pre-exposed subjects with clinical trial reports being limited in scope when addressing this issue [7,19,20,21]. Nevertheless, consistent with our findings presented here, a few reports suggest that antibody titers in previously infected individuals tend to be or are significantly higher than in SARS-CoV-2 naïve vaccinated persons [22,23,24,25].

## 2. Material and Methods

### 2.1. Cohorts

The samples in this study were derived from two main sources:

(1) From adult volunteers (>21 years old) participating in the IRB approved clinical protocol “Molecular Basis and Epidemiology of Viral infections circulating in Puerto Rico”, Pro0004333. Protocol was submitted to, and ethical approval was given by Advarra IRB on 21 April 2020. This is a running 5-year protocol which encompasses the collection of blood samples from adults exposed or suspected to be exposed to viral infections. An informed consent form and a study questionnaire also approved by the IRB were administered to the volunteers. From March 2020 to April 2021, we were able to follow up for serial samples with at least 59 subjects. From those 59, five subjects received two doses of Pfizer’s vaccine and two received Moderna’s formulation. We also added three vaccinated subjects for a total of 10 (ID511, ID512, and ID297). From those three, two received the Pfizer vaccine and one received the Moderna vaccine. All three subjects also consented to this study. In addition, a subgroup of 21 vaccinated volunteers, from the 59 subjects, that were never exposed to SARS-CoV-2 were followed for six to eight months (Appendix A). Of these 21 vaccinated volunteers, 18 received Pfizer’s vaccine and three received Moderna’s formulation. Those 21 subjects are part of the 59 subjects followed for months. During the follow-up period before vaccination, they never had symptoms or a positive serologic result. (2) De-identified blood samples were received from local laboratories network and blood banks. These subjects were self-enrolled for the purpose of donating plasma for the treatment of COVID-19 patients. Subjects were verbally informed regarding the relevance of their participation in COVID-19 research and were informed of the possibility that their deidentified samples may be used for research purposes. Subjects were given the opportunity to ask questions of blood bank workers regarding their participation. Furthermore, collected samples were handled using the standard blood donors’ protocols, and were accompanied by the blood bank’s signed consent form, which also detailed the possibility that samples would be used for research purposes. In addition, prior to receipt, samples were stripped of all identifiers so that the information cannot be traced back to the individual.

In summary, 134 samples were collected from the 59 subjects prior vaccination, while 38 and 99 serial samples were collected from the naturally infected group (10) and healthy cohort (21) receiving the vaccines, respectively.

As expected, some of the exposed subjects had more symptoms than others, with fever and loss of smell and taste being the most common symptoms. However, in this cohort, subjects did not need hospitalization or additional medical support in an emergency room setting.

### 2.2. Detection of SARS-CoV-2 IgM Antibodies

CovIgM-Assay is an indirect ELISA for the determination of human IgM antibody class, which was optimized via checkerboard titration. This assay is a laboratory-developed test (LDT) with an emergency use authorization (EUA) submitted to the U.S. Federal Drug Administration (FDA) (EUA202043). In summary, microplates were coated overnight at 4 °C with 2μg/mL of recombinant SARS-CoV-2 S1-RBD/S2 protein (GenScript No. Z03483-1) in carbonate–bicarbonate buffer. Plates were washed three times with phosphate buffered saline (PBS) containing 0.05% Tween-20 (PBST) and blocked for 30 min (min) at 37 °C with 250 μL/well of 3% Bovine Serum Albumin (BSA) in PBST. Diluted serum or plasma samples (1:100 in blocking buffer) were added in duplicates to the wells and incubated at 37 °C for 30 min. The excess antibody was washed off with PBST. Horseradish peroxidase (HRP) labeled-mouse anti-human IgM-mu chain (Abcam) diluted 1:30,000 in PBST was added (100 μL/well) and incubated for 30 min at 37 °C. After another washing step, TMB solution was added (100 μL/well) followed by a 15 min incubation. The reaction was stopped by the addition of 50 μL/well 10% HCl and the absorbance was measured at 450 nm (A450) using a Multiskan FC reader (Thermo Fisher Scientific). In every CovIgM assay determination, four wells in which samples were replaced by 100 μL/well of PBST were included as background control. Moreover, two in-house controls, a high positive control (HPC) and negative control (NC), were included. HPC and NC were prepared by diluting an IgM anti-SARS-CoV-2 at a concentration of 80 μg/mL and 0.070 μg/mL, respectively, in PBST containing 10% glycerol. The IgM anti-SARS-CoV-2 was purified from the plasma of a convalescent patient using 5/5 HiTrap IgM columns (GE Healthcare, USA). When the OD value of a serum or plasma sample at the working dilution (1:100) was equal or less than the cut-point (OD450 = 0.229), the CovIgM assay in the sample was assumed to be negative.

### 2.3. Detection of SARS-CoV-2 IgG Antibodies

IgG antibodies were detected and quantified using the CovIgG assay [26]. This assay is a laboratory-developed test (LDT) with an emergency use authorization (EUA) submitted to the U.S. Federal Drug Administration (FDA) (EUA201115). It is an indirect ELISA for quantitative determination of human IgG antibody class, which was optimized by checkerboard titration. In summary, disposable high bind flat-bottomed polystyrene 96-wells microtiter plates (Costar, Corning MA No. 3361) were coated overnight at 4 °C with 2μg/mL of recombinant SARS-CoV-2 S1-RBD/S2 protein (GenScript No. Z03483-1) in carbonate–bicarbonate buffer (Sigma Aldrich No. 08058). Plates were washed 3 times with (PBST) and blocked for 30 min at 37 °C with 250 μL/well of 3% non-fat, skim milk in PBST. Samples (serum or plasma) were diluted 1:100 in PBST; 100 μL/well was added in duplicates and incubated at 37 °C for 30 min. The excess antibody was washed off with PBST. Horseradish peroxidase (HRP) labeled-mouse anti-human IgG-Fc specific (GenScript No. A01854) diluted 1:10,000 in PBST was added (100 μL/well) and incubated for 30 min at 37 °C. After another washing step, a substrate solution (Sigma Aldrich No. P4809) was added (100 μL/well) followed by 15 min incubation. The reaction was stopped with 50 μL/well 10% HCl and the absorbance was measured at 492 nm (A_492_) using a Multiskan FC reader (Thermo Fisher Scientific). In every CovIgG-Assay determination two in-house controls, a HPC and NC were included. HPC and NC were prepared by diluting an IgG anti-SARS-CoV-2 at a concentration of 30 μg/mL and 0.070 μg/mL, respectively, in PBST containing 10% glycerol. The IgG anti-SARS-CoV-2 was purified from plasma of a convalescent patient using a 5/5 HiTrap rProtein-A column (GE Healthcare, NJ, USA). When the OD value of a serum or plasma sample at the working dilution (1:100) was equal or less than the cutoff-point (OD492 = 0.312), the CovIgG-Assay in the sample was assumed to be negative. However only samples with OD above of 0.499 were reported as having a titer within a range of 1:100 to ≥1:12,800.

For isotyping ELISAs, the conjugate was changed for the specific isotype as follows: anti-IgA (alpha chain specific-HRP (Sigma); anti-IgG1, 2, 3, and 4; and Fc-specific-HRP (Southern Biotech). All conjugates were used in a 1:3000 dilution.

### 2.4. cPass SARS-CoV-2 Neutralization Antibody Detection Method

To determine the neutralizing activity of antibodies, we used a surrogate viral neutralization test (C-Pass GenScript sVNT, Piscataway NJ, USA) [17,18]. Briefly, serum or plasma samples were diluted according to manufacturer’s instructions and incubated with soluble SARS-CoV-2 receptor-binding domain (RBD-HRP) antigen for 30 min, mimicking a neutralization reaction. Following incubation, samples were added to a 96-well plate coated with human ACE-2 protein. RBD-HRP complexed with antibodies were removed in a wash step. The reaction was developed with tetramethylbenzidine (TMB) followed by a stop solution allowing the visualization of bound RBD-HRP to the ACE2. Since this is an inhibition assay, color intensity was inversely proportional to the number of neutralizing antibodies present in samples. Data were interpreted by calculating the percent of inhibition of RBD-HRP binding. Samples with neutralization activity of ≥ 30% indicated the presence of SARS CoV-2 RBD-interacting antibodies capable of blocking the RBD–ACE2 interaction, thus inhibiting viral entry into host cells. While this assay measures the blocking activity of those antibodies, this activity is referred to throughout the text as “percentage of neutralization” for consistency and clarity.

### 2.5. Focus Reduction Neutralization Test

A focus reduction neutralization test (FRNT) was used to quantify the titer of neutralizing antibodies against SARS-CoV-2 isolate USA-WA1/2020. For this assay, we selected 41 samples, pre-vaccination, out of the 134 from the cohort of 59 subjects.

This assay was performed as described by our group and our collaborators before [27,28]. In summary, Vero WHO cells were plated in a 96-well flat-bottom tissue culture-treated plate. For determination of titer of a sample or virus stock, serial 10-fold dilutions of the sample were made in a 96-well round-bottom plate containing 5% DMEM. Plates containing sample dilution incubated with 37 °C, 5% CO_2_ for 1 h. On the day of the assay, 2% methylcellulose was diluted 1:1 in 5% DMEM (overlay) and placed on a rocker to mix. After the one-hour infection period, the 96-well plates containing sample dilution and cell monolayer were removed from incubator. Overlay was added to the plate by adding 125 μL of overlay media to each well and plates were incubated with 37 °C, 5% CO_2_ for 24 h. Following the 24-h incubation, the 96-well plates were fixed in a solution of 5% paraformaldehyde (PFA) diluted in tissue culture grade 1 Å~ PBS. Plates were removed from the incubator and the media containing the overlay and sample were aspirated off. One wash with 150 μL of 1 Å~ PBS per well was performed and 50 μL per well of 5% PFA in PBS was added for the fixing step. After 15 min in 5% formalin-buffered phosphate (Fisher: SF100-4), the plates were removed from the formaldehyde bath and the 5% PFA was removed from the monolayer. One wash with 100 μL of 1 Å~ PBS (tissue culture grade) per well was performed. The plates were submerged in a bath of 1 Å~ PBS to rinse and remove from BSL-3 containment. Foci were visualized by an immunostaining protocol. The 96-well plates were first washed twice with 150 μL per well of FFA wash buffer (1 Å~ PBS, 0.05% Triton X-100). The primary antibody consisted of polyclonal anti-SARS-CoV-2 guinea pig sera (BEI: NR-0361) and was diluted 1:15,000 with FFA Staining Buffer (1 Å~ PBS, 1 mg/mL saponin (Sigma: 47036). Then, 50 μL per well of primary antibody was allowed to incubate for 2 h at room temperature or 4 °C overnight. The 96-well plates were then washed three times with 150 μL per well of FFA Wash Buffer. The secondary antibody consisted of goat anti-mouse conjugated horseradish peroxidase (Sigma: A-7289) diluted 1:5000 in FFA Staining Buffer. Similarly, 50 μL per well of secondary antibody was allowed to incubate for 2 h at room temperature or 4 °C overnight. The plates were washed three times 497 with 150 uL per well of FFA wash buffer. Finally, 50 μL per well KPL Trueblue HRP substrate was added to each well and allowed to develop in the dark for 10–15 min, or until blue foci are visible. The reaction was then quenched by two washes with Millipore water and imaged immediately thereafter with a CTL machine to quantify foci.

### 2.6. Statistical Methods

Statistical analyses were performed using GraphPad Prism 7.0 software (GraphPad Software, San Diego, CA, USA). The statistical significance between or within groups evaluated at different time points was determined using two-way analysis of variance (ANOVA), one-way ANOVA (Tukey’s, Sidak’s, or Dunnett’s multiple comparisons test as post-hoc test), unpaired t-test, or Wilcoxon–Mann–Whitney, to compare the means. The *p* values are expressed in relational terms with the alpha values. The significance threshold for all analyses was set at 0.05; *p* values less than 0.01 are expressed as *p* < 0.01, while *p* values less than 0.001 are expressed as *p* < 0.001. Similarly, values less than 0.005 are expressed as *p* < 0.005. Cohen’s Kappa agreement follow Landis and Koch scale. The values (κ) were considered as follows: poor agreement, κ < 0.2; fair agreement, κ = 0.21 to 0.4; moderate agreement, κ = 0.41 to 0.6; substantial agreement, κ = 0.61 to 0.8; and very good agreement, κ = 0.81 to 1.0.

## 3. Results

### 3.1. Sample Collection

Subjects were enrolled and samples were collected as participants became willing and available. However, the time between serial samples was very similar for all subjects. The average time between the moment of the documented infection and the first collected samples (*n* = 59) was 40.37 days (minimum 12 days, maximum 97 days and two extreme cases with 127 and 176 days for a median of 38 days). Once the subjects entered in the cohort, the average time between the first and the second samples (*n* = 59) was 67.86 days (minimum 7 days, maximum 111 days, median 67.5 days). The average time between the second and the third samples (*n* = 12) was 99.5 days (minimum 63 days, maximum 159 days, median 95 days). The single fourth sample was collected 3 months after the third samples (Appendix A).

From the two subgroups (exposed vaccinated and unexposed vaccinated individuals) serum samples were collected between 15 to 20 days after each dose. In addition, a third sample from all 21 unexposed individuals and from eight out of the 10 pre-exposed participants was collected between 19 and 83 days after the second dose (average of 40.1 and of 81.6 days for the unexposed and pre-exposed groups, respectively) or an average of 60.3 and 100.5 days after the first vaccine dose for the unexposed and pre-exposed groups, respectively (Appendix A). It is highly relevant for our findings that the sample used as baseline in the pre-exposed individuals before vaccination was collected in an average of 142 days after the confirmed infection (minimum 67 days, maximum 310 days, median 126.4 days) (Appendix A).

### 3.2. SARS-CoV-2 Specific IgG Titers Decline over Time

Overall, the IgG titers in the cohort of 59 subjects were significantly higher (geometric mean 1072) in the first set of samples than the second set of samples (geometric mean 618) (*p* < 0.0473) or the third set of samples (geometric mean 537) (*p* < 0.0474). We observed no significant differences between titers measured in the second and third sets of samples (*p* < 0.3085) (Figure 1A). The results are reported as OD450 in Appendix A and agree with estimated titers (Appendix A).

Of the 59 subjects naturally exposed to the virus, 40 (67.8%) experienced a decrease in IgG titers (Figure 1B) while 19 (32.2%) showed an increase in the IgG titers from the first to the second set of samples (Figure 1C). We found no relationship between the elapsed time from initial diagnosis to first sample collection and the change (e.g., increase or decrease) in IgG titers between sample collections (Appendix A). From these results, we concluded that the differences in the IgG titers in those groups from the first to the second set of samples were not attributable to the time between collection. We also found no relationship between the elapsed time between the first and second sample collection for both groups (Appendix A).

### 3.3. SARS-CoV-2 Specific IgM Titers Decline over Time

Among the 59 subjects in our cohort, 37 (62.71%) had detectable IgM titers in the first set of samples, while 18 (30.50%) had detectable IgM titers in the second set of samples (Appendix A). In five subjects out of the 12 where a third sample was collected, IgM titers were still detectable. In some cases, subjects developed an IgM response for first time (in volunteer ID313 IgM was detected as early as 12 days after the presumptive diagnosis and persisted up to 192 days, or roughly 6.4 months). Overall, IgM titers showed a consistent pattern of decline in the second sample for most individuals (86.44%). Only one subject (ID265) showed no appreciable change in IgM titers between the first and second sample collection (68 days elapsed). One subject (ID313) displayed no measurable IgM titer at the time of the first and second sampling (106 days elapsed) but appeared positive for IgM titers in the third sample (146 days elapsed). We also found that, in the second set of samples, three subjects out of 59 (5.08%) displayed detectable IgM titers which were absent detectable IgG titers. Subject ID312 showed detectable IgM titers, but borderline IgG titers results in the third sample collected 57 and 69 days after the first and second dose, respectively (86 and 98 days after the presumptive diagnosis). Subject ID105 still had detectable IgM titers 192 days after the presumptive diagnosis was made. The earliest time point with detectable IgM titers was 12 days after the presumptive infection (ID166), followed by 13 days (ID180) and 14 days (ID179) after diagnosis. In general, IgM was detected in 37 subjects (77.97%) in the first set of samples (43 days post presumptive infection). In 18 subjects (57.63%), IgM was detected in the first and second set of samples (104 days post presumptive infection). In four subjects (6.77%), no IgM was detected in any of the serial samples collected.

### 3.4. IgG Titers—But Not IgM or IgA Titers—Correlate with Neutralizing Activity

We first aimed to compare the performance of the surrogate assay cPass SARS-CoV-2 neutralization antibody detection method selected for this work with the classical FRNT. Our results indicate that the capacity of two different assays to detect positive samples was of 95.12% (Cohen’s Kappa value of 0.474 for a moderate agreement) (Appendix A). Then, we performed the same analysis by examining the correlation between IgG titers and their functional neutralization capacity. By applying a Kappa analysis, we first aimed to determine if both techniques agree when classifying positive and negative samples using a titer ≤100 and >30% as a cutoff for the IgG titers and a percentage of neutralization, respectively. We found moderate agreement between IgG titer and neutralization capacity, with a Cohen’s Kappa value of 0.4304 (Appendix A). We then aimed to determine whether both techniques agree when classifying samples with high IgG and neutralizing antibody titers. Similarly, we found moderate agreement between IgG titer and neutralization capacity, with a Cohen’s Kappa value of 0.5402 (Appendix A). We completed the same analysis for IgM and IgA titers to explore the contribution of those antibody subclasses to total neutralization capacity. We found that both techniques (IgM titer and cPass) have a fair agreement when classifying positive and negative samples (Cohen’s Kappa = 0.2391), while the IgA titer and the neutralization assay showed only a slight agreement (Cohen’s Kappa = 0.0618) (Appendix A).

### 3.5. Neutralizing Activity Remains Constant over Time

To determine the durability of the neutralizing antibody response, we examined the neutralization capacity in our longitudinally collected samples. Our results showed consistent neutralizing antibody titers over time, with no change in the neutralization potential from the first (geometric mean 68.08%) to the second (geometric mean 63.89%) sample. Similarly, we saw no appreciable decline in neutralization potential from the second (geometric mean 63.89%) to the third (geometric mean 60.36%) sample (Figure 1D and Appendix A). We did, however, identify two distinct trends in the kinetics of serum neutralization potential over time. Similar to our findings with total IgG titers, in the first collected sample, we found a decrease in the neutralizing activity relative to the second sample in 61.01% (36 out of 59) of the subjects. Conversely, 38.98% of subjects (23 out of 59) showed a decrease in neutralization activity (Figure 1E,F) during the same timeframe. While the percentage of subjects experiencing an increase or a decrease in neutralization capacity and IgG titers between samples was similar, the change in neutralization capacity was less pronounced and not significant compared with significant changes in the IgG titers (Figure 1B). From these findings, we concluded that the neutralizing capacity remains relatively constant during the time we followed this cohort.

Similarly, we compared the neutralization potential of sera from subjects in the second and the third samples for the few subjects (*n* = 3) for which we were able to obtain a third sample. We identified one subject (ID313) showing a different pattern, with a 3.34-fold (68%) increase in neutralizing activity from the second to the third sample. Another two subjects showed an increase in IgG titers but displayed a very limited increase in neutralizing activity of 1.2-fold (ID135) and 0-fold (ID195). Despite the variability in IgG titers, neutralizing activity remained over 50% in a majority (90%) of all three samples. The distinctive serological and neutralization pattern for subject ID313 appears to be strongly related to the clinical evolution (Appendix A).

We also identified 11 subjects without detectable SARS-CoV-2-specific IgG titers which showed some degree of neutralization ranging from 36% to 76%. Six out of those 11 subjects had no detectable total IgG. On the other hand, there were three subjects with detectable IgG titers capable of binding SARS-CoV-2 S protein, but with very limited or absent neutralization capacity (Appendix A).

### 3.6. Natural Infection Induces High Quality Antibodies Than One Vaccine Dose

Next, we wanted to compare the magnitude of the humoral immune response to naturally acquired SARS-CoV-2 infection to the mRNA-based COVID-19 vaccinations in unexposed subjects. For this purpose, we chose samples from 25 participants out of the 59 with the first sample collected between 12 and 38 days after the confirmed infection with SARS-CoV- (average 25.72 days, min 12, max 38) and from 21 unexposed participants that received two doses of the Pfizer-BioNTech vaccine (average 17.35, min 12, max 26). Samples for the unexposed subjects were collected over an average of 17.1 and 14.1 days after the first and the second dose, respectively. As shown in Figure 2A, the mean time elapsed between the first sample collection after infection was significantly higher than the time elapsed between the first sample collected after the vaccination in the unexposed cohort (*p <* 0.0001). Despite this delay, we found that the total anti-S antibodies and the total IgG titers were comparable after the infection or the first vaccine dose in the unexposed participants (Figure 2B,D). However, the quality of the antibodies measured by the surrogate neutralization assay showed a neutralizing activity significantly higher in the naturally infected group compared with the unexposed vaccinated group (*p* < 0.0003). This indicated a better quality of the antibodies induced by naturally acquired infection when compared to vaccine-induced neutralizing antibody activity (Figure 2D). As shown in Figure 2B,C, two vaccine doses in unexposed individuals were necessary to significantly increase the total antibody titers and IgG titers compared to individuals in the pre-exposed group (*p* < 0.0001). The magnitude of neutralization was also significantly increased in pre-exposed individuals (*p* < 0.0001) (Figure 2D).

### 3.7. Neutralization Is Sustained in Naïve and Pre-Exposed-Vaccinated Subjects

Samples were collected between 12 to 28 days after each dose with a mean of 19 days and of 14 days for the pre-exposed group and of 12 days and 26 days for the unexposed groups after the first and second dose, respectively. An additional third sample from all 21 unexposed individuals and from 8 out of the 10 pre-exposed individuals was collected between 19 and 83 days after the second dose, respectively (Appendix A). For the first sample collected following the first dose, there were no significant differences in the time elapsed between sample collections for the pre-exposed and unexposed subjects. However, there was a significant difference (*p <* 0.0001) in the time elapsed between sample collections following the second dose (third sample) between the pre-exposed and unexposed groups (Appendix A). The geometric mean baseline IgG titers in the pre-exposed population was 726 (range: 125 to 7191) and increased to a geometric mean of 5239 (range: 3408 to 6586) after the first dose (Figure 3B and Appendix A). After the second dose, the geometric mean decreased to 3980 (range: 2273 to 5847), and we observed no significant difference in IgG titers after the first dose. On the other hand, the 21 vaccinated, unexposed subjects were negative for S-specific IgG at baseline. After the first dose, the IgG titers significantly increased to a geometric mean of 832 (range: 196 to 9365, *p* < 0.0001) and, after the second dose, those values significantly increased (*p* < 0.0001) to a geometric mean of 5446 (range: 3346 to 10,239) (Figure 3B).

In the second sample, which was collected after the second dose (third sample) in the unexposed group, the geometric mean of the titers was 1518 (range: 409 to 3278). In the pre-exposed group, the geometric mean of the titers was 1323 (range: 568 to 3536). In both groups, we observed a significant decrease from the IgG titers detected in the first samples relative to titers after the second dose (*p <* 0.0001 and *p* = 0.0192 for the unexposed and pre-exposed groups, respectively).

In our cohort, the total IgG values were consistent with reported IgG titers (Figure 3A). We looked first at the IgG1 isotype, the main contributor to the total IgG in the cohort of 59 individuals. The first dose induced a significant increase in this isotype for both groups (*p* < 0.0018 and *p* < 0.0001 for the unexposed and pre-exposed vaccinated groups, respectively). However, the effect of the boost was significantly higher in the pre-exposed group (*p* < 0.0001), suggesting a role for natural infection in this significant difference. Remarkably, the second dose appeared to provide a benefit in boosting IgG1 titers in the unexposed vaccinated group only (*p* < 0.0001). IgG1 values after the second dose in the unexposed vaccinated group reached values comparable to that of the pre-exposed vaccinated group after just one dose. We observed no significant differences in the levels of IgG1 between groups following the second dose (Appendix A).

The geometric mean baseline of neutralization activity in the pre-exposed population was 69.46% (range: 39 to 97%) and increased significantly (*p* < 0.0001) to a geometric mean of 97.99% (range: 97 to 98%) after the first dose (Figure 3C and Appendix A). However, following the second dose, the values remained similar in range, with a mean of 97.19%. On the other hand, the 21 naïve vaccinated persons were negative for neutralization at baseline (geometric mean: 15%). After the first dose, neutralization significantly increased (*p* < 0.0001) to a geometric mean of 57.34% (range: 28% to 76%, with one outlier of 96%). The second dose produced an additional significant boost (*p* < 0.0001) to a geometric mean of 96.85% (in a range from 95% to 98%) (Figure 3C). Contrary to the trend we observed in total antibody titers and IgG titers(Figure 3A,B), the neutralizing activity was retained at very similar level in both groups in the third sample collected. The geometric mean for the unexposed group was 94.5% (in a range from 86% to 98%), while the pre-exposed group had a geometric mean of 96.62% (in a range from 96% to 98%). Though there was no significant difference in neutralization capacity between groups, nine (9) subjects in the unexposed group showed values lower than 5% neutralization. This resulted in a 1.02-fold decrease in the value of neutralization capacity in the unexposed group, while there were no changes in neutralization capacity the pre-exposed cohort.

Among the previously exposed subjects we examined, 5 out of 10 (50%) retained detectable IgM at baseline (i.e., the time of the first sampling). IgM titers did not appear to be boosted by the first vaccine dose, and titers decreased after the second dose. On the other hand, the first dose did appear to induce a significant increase (*p* < 0.0001) in the IgM values in the unexposed subjects. Those values were boosted only in two subjects, but as expected, were not modified in any of the other 19 subjects (Appendix A). Eight (8) out of the 21 unexposed patients (38.09%) had no detectable IgM after the first dose. Only one patient failed to develop measurable IgM antibodies after the two vaccine doses.

Finally, we looked at the contribution of the IgA isotype to the immune response after vaccination. Interestingly, we found that this isotype was significantly boosted in both groups, pre-exposed (*p* < 0.0187) and unexposed groups (*p* < 0.0010) after the first vaccine dose. In addition, the increase in IgA titers was significantly higher in the pre-exposed (*p* < 0.0176) vaccinated group compared to the unexposed vaccinated group. The second boost resulted in an additional significant increase in IgA titers in the unexposed vaccinated population but not in the pre-exposed vaccinated group (Appendix A).

## 4. Discussion

Our study followed a cohort of 59 subjects with prior exposure to SARS-CoV-2 with the goal of describing the kinetics of the humoral immune response to natural infection over time. This study uniquely examined a population of Hispanic/Latino persons disproportionately impacted by the COVID-19 pandemic. We compared the kinetics of this antibody response in the context of individuals with naturally acquired infection (pre-exposed) and unexposed individuals following vaccination. None of the exposed subjects in our cohorts required hospitalization and only had mild to moderate symptoms. Because of that, we found no differences in the serological response according to symptoms severity. Consistent with other reports, we found that antibody titers tended to wane over time and added to a growing body of evidence suggesting that functional neutralization assays should serve as the gold standard for evaluating vaccine efficacy in lieu of antibody binding quantification. Furthermore, we found that pre-exposed individuals were able to mount an antibody response after just one vaccination dose that was equivalent to a two-vaccine dose regiment in unexposed individuals. These findings have important implications for defining the correlates of protection for SARS-CoV-2, as well as recommendations for future public health guidelines and vaccine distribution efforts on a global scale.

One limitation of our work is the number of subjects sampled following natural infection or vaccination. However, we were able to draw statistically significant conclusions from our studies using 59 individuals. Additionally, our findings in this limited dataset are consistent with previous reports, which have made great contributions to our understanding of the immunological response to SARS-CoV-2 with a similar number of subjects [5,22,24,25].

We also acknowledge that setting up a longitudinal cohort study is always a challenge. Particularly for COVID-19, it imposed additional difficulties due to the lockdowns, social distancing measures, stigma associated with positive testing and other significant barriers. However, we assert that the limitations regarding the sampling sequence do not detract from the significance of our findings.

Notably, our results contrast with reports describing a short persistence of neutralizing antibodies in plasma donors [29], but are in agreement with recent work indicating that neutralizing antibodies may persist longer [4,9,30]. Another work showed a long-term stabilization of anti-S IgG values and nAbs which were lower than in early days post symptoms onset in a hospitalized cohort [31]. The effect we are seeing in the samples with a decrease in the total antibodies and titers in the second sample may be also a stabilization at a plateau. We have followed up samples from 8 out of the 10 pre-exposed vaccinated subjects but, unfortunately, alterations in the humoral response due to vaccination of these subjects limit our interpretation of these results. Interestingly, the same group reported that nAbs are a correlate of survival and that nAbs and anti-S IgG persist in the vast majority of recovered patients regardless of disease severity, age, and co-morbidities for up to eight months from symptoms onset [31]. A longer follow-up period would further our understanding of the antibody kinetics in a long-term period

We were able to show a similar trend in our cohort, with sustained neutralizing activity during the frame time of this study. The sustained neutralization capacity we observed remains highly relevant, despite the significant decline in IgG titers that we observed in this cohort. In addition, we found that some subjects with undetectable IgG (*n* = 6) and IgG titers (*n* = 11) retain measurable neutralization activity, ranging from 32 to 76%, as measured by a surrogate virus neutralization assay. This finding is consistent with previous reports, suggesting that SARS-CoV-2 serological assays may be poorly suited for prediction of serum neutralization potency, a metric necessary to facilitate the establishment of the appropriate serologic correlates of protection against SARS-CoV-2 [32]. Our results suggest that functional assays measuring neutralization potential should be implemented in studies of vaccine efficacy at the population level.

From a technical point of view, the discrepancies between samples without detectable antibodies but with neutralizing capabilities may be explained by differences in the assays’ sensitivity. In our case, we use the same source of recombinant proteins for the antibodies and surrogate neutralization assays. However, the serological assays include the full S1 and S2 regions of the S protein, which includes the RBD, to coat the plate. The neutralization assay, on the other hand, includes only the S1/RBD in suspension. It has been well documented that the binding of the protein to the plate results in altered antigen accessibility with a consequent presentation of different antigenic sites compared to native proteins [18,33,34,35]. Nevertheless, we showed a 93.7% correlation between IgG titers and neutralization measured with a cPass SARS-CoV-2 Neutralizing Antibody Detection kit.

There are a limited number of publications on the contribution of different antibody isotypes to the immune response to this novel coronavirus. Early studies reported that S- and RBD-specific IgM, IgG1, and IgA antibodies were detected in most subjects early after infection, with all samples displaying neutralizing activity and IgM and IgG1 contributing most to neutralization [30]. A recent work reported that, in a hospitalized cohort, the early presence of anti-RBD anti-S IgA positively correlated with reduced persistence of SARS-CoV-2 RNA in naso-pharyngeal swabs [31]. Other work reported that early SARS-CoV-2-specific humoral responses were dominated by IgA antibodies and that virus-specific antibody responses included IgG, IgM, and IgA. Furthermore, some studies have found that the IgA isotype contributes to virus neutralization to a greater extent compared with IgG [36]. In agreement with our results, recent work from India, a heavily impacted country by the pandemic, found that RBD-specific IgG, but not IgA or IgM titers, correlated with neutralizing antibody titers and RBD-specific memory B cell frequencies [37]. In our work, we found that IgG1 was the predominant isotype, while the IgA response was more limited. However, considering the non-significant changes in the IgA levels from the first to the second sample, a role for IgA in sustained neutralization activity cannot be ruled out. On the other hand, in most of the subjects in this cohort, an expected trend of IgM decline was observed in the second collected sample. Two out of four subjects (ID265 and ID382), which were IgG-/IgM+, also had detectable neutralizing activity with detectable IgM both two and four months after the first samples were collected. These cases suggest that, in some individuals, IgM may contribute to sustained neutralization capacity, as has been described before [30]. This result also corresponds with a Kappa analysis suggesting a fair Cohen’s Kappa agreement between IgM titers and neutralization capacity. Additional isotype-specific depletion experiments are needed to determine the role of these antibodies in SARS-CoV-2 neutralization. Using previous experience from our group [28,38], those experiments are underway using a larger number of well characterized individuals.

While the number of subjects in our vaccinated cohort (both unexposed and previously exposed subjects) is limited, we show that vaccination induces a higher boost in the magnitude of the humoral immune response, both at the level of S-specific IgG and neutralization ability in the pre-exposed individuals compared to the naïve group. Our findings also indicate that the second vaccine dose did not expand the S-specific antibodies, the total IgG titers, or the neutralization capacity of blocking antibodies beyond the peak reached after the first dose in the case of the pre-exposed cohort. One subject (ID112) who received the Moderna formulation (ID112) was identified as unexposed and without any known exposure to the SARS-CoV-2, reaching values in all three determinations comparable to that of the pre-exposed subjects. Notably, however, for volunteers who worked in a high-risk environment during the first months of the pandemic, asymptomatic infection cannot be ruled out despite the absence of measurable S-specific and neutralizing antibody titers at baseline.

Our study revealed two significant findings regarding vaccination. First is the rapid decline in anti-S antibodies just 40 to 80 days (for unexposed or pre-exposed cohorts, respectively) after a boost with the mRNA vaccine formulations. Second is the sustained level of neutralization ability in the same period that anti-S antibodies are declining. This pattern is the same as the one observed following naturally acquired SARS-CoV-2 infection in 59 subjects. In addition, we observed that—while in both groups the decline in the total anti-S antibodies and IgG titers was significant—the decline in titers was more precipitous in the unexposed group relative to the pre-exposed group. Furthermore, highly significant is the observation that the baseline neutralizing activity—but not the total antibody titers—was significantly higher among pre-exposed individuals than the neutralization capacity induced by the first vaccine dose in the unexposed group. This finding is reinforced by the fact that the time after natural infection and the sample used as baseline before vaccination was more than 4.7 months in average for all 10 pre-exposed subjects. Our results also confirm that antibodies generated after the natural infection, while similar in quantity, are significantly better in their function when natural infection preceded vaccination. These results suggest that natural infection with SARS-CoV-2 may contribute to the expansion of memory B cells, enabling the production of more S-specific antibodies following vaccination. Together, these findings highlight the value of measuring both the function and quantity of S-specific antibodies to follow up humoral immune responses to the vaccination. Our results agree with recent work wherein a predictive model of immune protection from COVID-19 found that the level of neutralizing antibodies is highly predictive of immune protection from symptomatic SARS-CoV-2 infection [23] and associated to recovery [31].

Our results on neutralization are built on using the RBD sequence from the authentic SARS-CoV-2 virus. Some works demonstrated the presence of non-RBD targeting antibodies possessing neutralizing capacity [39,40] that may not be detected by the surrogate assay we implemented. We may be missing the contribution of some non-RBD neutralizing antibodies, however, the RBD domain continues to be the key target for SARS-CoV-2 neutralization [41,42,43,44,45]. Any other additional neutralizing activity we may be missing would result in an increase in the antibodies function, reinforcing the effect we are reporting here, where function is more reliable than the presence or not of antibodies to follow up the immune response to this novel coronavirus and to the COVID-19 vaccines. On the other hand, there are disperse reports showing the detection of non-neutralizing antibodies by sVNTs. We cannot rule out the contribution of those non-neutralizing/non-binding antibodies to our results. However, from 41 samples tested with both assays, we found only 3 samples with FRNT+/sVNT-, suggesting the presence of some non-RBD neutralizing antibodies. However, we did not identify any sample with the opposed profile FRNT-/sVNT+. The effectiveness of this assay is continuously reported in the literature, confirming its high specificity [46,47,48,49,50,51]. As a result, we consider that, under our experimental conditions, the contribution of non-neutralizing antibodies to our results can be considered very limited.

Any variants infecting the subjects remain undetermined. However, all 59 subjects in the serial samples’ cohort were exposed to the SARS-CoV-2 from March to December 2020. Only the three additional subjects in the pre-exposed and vaccinated cohorts were confirmed as positive during the first two weeks of January 2021. During that period, information about the circulating variants in Puerto Rico was very limited. The first variant identified in Puerto Rico was the Alpha variant (first identified in the UK, B.1.1.7) and was reported on 28 January 2021. In addition, from March 2020 to December 2020, the government of Puerto Rico imposed a strict lockdown, limiting the travels to the island and enforcing mandatory testing upon arrival. By 21 July 2021, reports from the surveillance system from the PR Department of Health and other private institutions reported about 950 cases, with patients infected with at least nine (9) different variants as follows: UK Alpha (B.1.1.7), New York (B.1.526), Brazil Gamma (*p*.1), California Epsilon (B.1.429) and (B.1.427), California Eta (B.1.525), India Delta (B.1.617), Brazil Zeta (P.2), South Africa Beta (B.1,351), and India Kappa (B.1.617). It is important to highlight that the pattern of neutralizing antibodies is very dynamic and can only be interpreted at the individual level [52]. We acknowledge that the neutralizing properties of our samples may be modified when tested against the RBD from the variant of interest and variant of concerns. However, a work testing four variants representing the original SARS-CoV-2 strain and emerging variants with mutations in the S protein suggested that infection- and vaccine-induced immunity may be retained against the B.1.1.7 variant [53].

Of interest is the role of previous natural infection in driving antibody isotype switching. Particularly in the case of IgA, our results showed that previous exposure led to a faster increase in IgA titers after the first dose of vaccination, while unexposed subjects required a second dose of vaccine to reach same levels of IgA titer of those pre-exposed to the novel coronavirus.

Another critical aspect to be considered is the timing between the natural infection and a potential vaccination against COVID-19. In accordance with the findings of other groups, we highlighted the relevance of the time elapsed between infections or immunizations to induce an optimal immune response [38,54,55]. Taking into account the results presented here and those from previous works [22,25,56], and considering the limited vaccine availability worldwide, our findings suggest that immunity conferred by a single dose may be sufficient to provide immune protection from severe disease in previously exposed individuals. With this in mind, second doses in previously exposed immunocompetent individuals may be deferred until the final phases of vaccination campaigns and/or to be executed not before than 6 months after the documented infection. In fact, recent results from Israel using a larger cohort reinforce our results [57].

Because of the limited number of samples, we were unable to identify any significant differences between the Pfizer-BioNTech or Moderna vaccine formulations.

We are aware of the limitations of this work owing to the limited number of participants and associated clinical data. We also understand that this work would benefit from an examination of the T-cell compartment in unexposed and pre-exposed vaccinees, particularly in light of recent evidence that simple serological tests for SARS-CoV-2 antibodies do not reflect the richness and durability of immune memory to SARS-CoV-2 [4]. With this in mind, experiments characterizing the T-cell response in our cohorts are underway.

Nevertheless, this work provides new and additional insight to the limited available data on COVID-19 immune phenomena. Furthermore, this work also advances our understanding of immune responses to the mRNA vaccine formulations in unexposed and pre-exposed individuals, outside of the data provided by the vaccine manufactures. From our results, as well as others [22,23,24,25], the usefulness of a second vaccine dose in pre-exposed subjects remains inconclusive. Furthermore, the immune response elicited by these vaccine formulations needs to be further evaluated to include the T cell compartment, which serves as a critical component in the response to SARS-CoV-2 [4,6,8,25]. Undoubtably, natural infection confers a strong and high-quality humoral and cellular immune response [4,6,57]. This fact has recently been underscored by work showing that variants of concern partially escape humoral—but not T-cell-mediated—immune responses in COVID-19 convalescent donors and vaccinees [5]. As the CDC’s guidelines and the impact of vaccination on our lifestyles (travel quarantine and testing, mask-less outside and indoors) continue to change and evolve, it remains unclear why immunity conferred by natural infection is not taken into account to support those guidelines, nor it is considered in the progress towards attaining herd-immunity that may enable us to return to the new social normality. In this context, our results are also highly relevant to consider standardizing methods that both serve as a tool to follow up the immune response to the vaccination, but also to provide a correlate of protection.

## Figures and Tables

**Figure 1 viruses-13-01972-f001:**
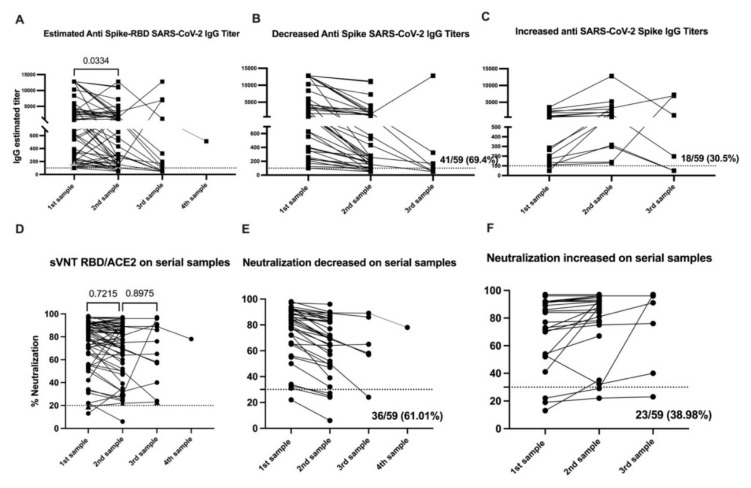
SARS-CoV-2 specific antibody titers decline over time, while neutralization ability is retained. Panel (**A**) shows the anti-Spike IgG titers, from all 59 samples, measured by enzyme-linked immunosorbent assay and expressed as titers. In panels (**B**,**C**), subset of samples with a pattern of decreased (*n* = 41) or increased titers (*n* = 18) in the second sample are presented. Panel (**D**) shows the blocking activity of serum antibodies expressed as percentage of neutralization by using a surrogate viral neutralization test (sVNT). In panels (**E**,**F**), subset of samples with a pattern of decreased (*n* = 36) or increased titers (*n* = 23) in the second sample are presented. The threshold for the total antibodies was 0.312. The threshold for IgG titers was 1:100 and for the blocking activity was 30%. The average time between the time of the documented infection and the first samples (*n* = 59) was 40.37 days, the average time between the first and the second samples (*n* = 59) was 67.86 days and the average time between the second and the third samples (*n* = 12) was 99.5 days. The single fourth sample was collected three months after the third sample. Statistical significance was determined by unpaired t-test or by one-way ANOVA multiple comparisons tests and were used to test for increase or decrease among samples. Tukey’s multiple comparisons test was performed as post-hoc test. *p* < 0.05 was considered significant. Samples 1 and 2 include the 59 subjects in the initial cohort before vaccination. Sample 3 encompass the 15 subjects from whom a collection of a third sample was completed.

**Figure 2 viruses-13-01972-f002:**
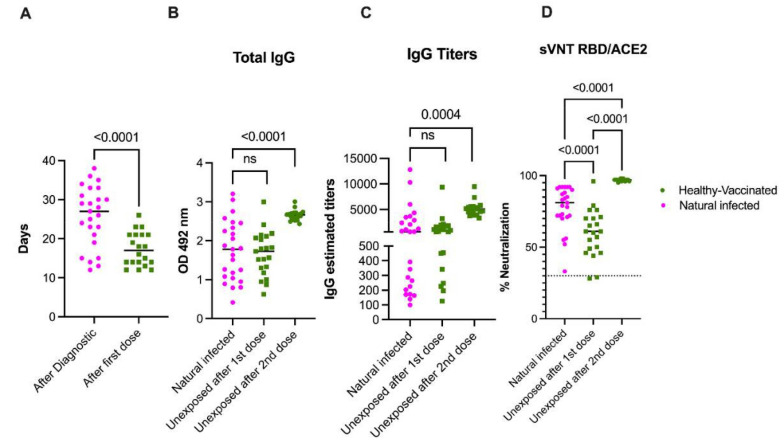
Naturally acquired SARS-CoV-2 infection primes an immune response superior to a single COVID-19 vaccine dose. Panel （**A**） shows the mean time of sample collection following natural infection (*n* = 25) or after the first vaccine dose (*n* = 20). In panels (**B**,**C**), results from the total anti-S protein and the IgG titer measured by enzyme-linked immunosorbent assay and expressed as OD or titers, respectively, are presented. The threshold for the total antibodies was 0.312 and the threshold for IgG titers was 1:100. All participants with previous exposure to SARS-CoV-2 except one showed detectable antibodies and measurable titers at baseline. Because the threshold 1:100 of our titration assay, the IgG titers at baseline in the unexposed subjects—which had no detectable S-specific antibodies—were set arbitrarily to 50. Panel (**D**) shows the blocking activity of serum antibodies expressed as percentage of neutralization by using a surrogate viral neutralization test (sVNT). The cutoff for this assay was 30%. As is shown, only one sample in the pre-exposed group contained antibodies below the threshold reported as more than 30% of neutralization. Furthermore, while the distribution of antibodies and titers covers the full Y axis, values in both panel B and C, and in panel D same samples are grouped on the top values area. Statistical significance was determined by one-way ANOVA multiple comparisons test, unpaired t-test or Wilcoxon–Mann–Whitney test, to compare the means and used to test for increases or decreases among samples. *p* < 0.05 was considered significant. Naturally infected participants (*n* = 25) out of the 59 with the first sample collected between 12 and 39 days after confirmed infection with SARS-CoV-2 were selected for comparison with the 21 unexposed vaccinated subgroup (healthy vaccinated).

**Figure 3 viruses-13-01972-f003:**
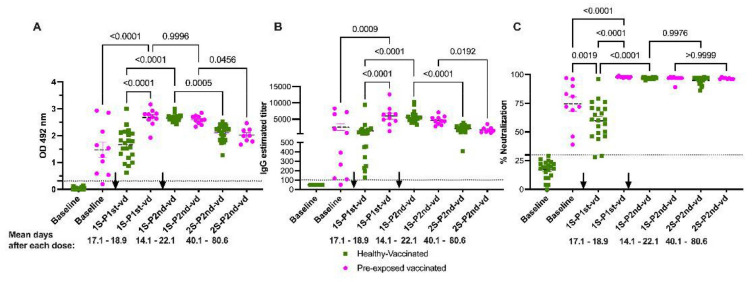
Functional neutralization assays are better predictors of the humoral immune response to COVID-19 mRNA-based vaccinations. Samples are described as the 1st or 2nd samples after 1st or 2nd vaccine dose (1 S-P1 st-vd, 1 S-P2 nd-vd or 2 S-P2 nd-vd) and the mean time of samples collection is shown. Panels (**A**,**B**) show the total antibody and IgG titers, respectively, after full vaccination with two vaccine doses. Antibody levels and titers significantly decline in both groups in a second sample collected after the second vaccine (average of 60.3 and 100.5 days after the first vaccine dose for the unexposed and pre-exposed groups, respectively). Despite the difference in sampling time between the two groups, there were no significant differences in the levels of antibodies or titers between groups in the 2 S-P2 nd-vd. Panel (**C**) shows antibody-blocking capabilities measured by a surrogate viral neutralization assay (sVNT). Highly relevant is the finding that the blocking baseline activity of the pre-exposed individuals is significantly higher than the basely blocking activity induced by the first vaccine dose in unexposed individuals. In addition, two vaccine doses were necessary in the unexposed cohort to induce same percentage of neutralization achieved by just the first dose in the pre-exposed group. The magnitude of neutralization remained at similar levels until the last time point evaluated in both groups, confirming that the surrogate neutralization test is more suitable to determine the efficacy of the humoral immune response to the vaccine. The threshold for the total antibodies was 0.312. The threshold for IgG titers was 1:100 and for the blocking activity was 30%. Statistical significance was determined by one-way ANOVA multiple comparisons test or unpaired t-test to test for an increase or decrease among samples. *p* < 0.05 was considered significant. The black arrows indicate the moment of vaccine administration related to the timing of sample collection. Healthy vaccinated (*n* = 21); pre-exposed vaccinated (*n* = 10).

## Data Availability

All data is available upon request.

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
