# Peer review of "Function Is More Reliable than Quantity to Follow Up the Humoral Response to the Receptor-Binding Domain of SARS-CoV-2-Spike Protein after Natural Infection or COVID-19 Vaccination"

_viruses, 2021, doi:10.3390/v13101972_

Round 1
Reviewer 1 Report
Title of the manuscript: Function is More Reliable than Quantity to Follow up the Hu- 2 moral Response to the Receptor Binding Domain of SARS- 3 CoV-2 Spike Protein after Natural Infection or COVID-19 Vac-cination
Manuscript ID: viruses-1371998
Evaluation Summary: This study by et al., followed a cohort of 59 subjects with prior exposure to SARS-CoV-2 with the goal of describing the kinetics of the humoral immune response to natural infection over time. Authors has shown that vaccination induces a higher boost in the magnitude of the humoral immune response, both at the level of S-specific IgG and neutralization ability in the pre-exposed individuals compared to the naïve group. Also, authors have noted rapid decline of anti-S antibodies just 40 to 80 days after boosting but have sustained levels of neutralization capacity.
Overall, the manuscript was well written, and discussion was up to the mark. Conclusions drawn from the analysis were clear. But I, have few concerns outlined below to be addressed by the authors:
Strengths of the study:
- a longitudinal cohort study of COVID-19 mRNA vaccinees.
- Very good discussion.
Weakness:
- As stated by the authors in discussion, a limitation of this work is the limited number of subjects sampled following natural infection or vaccination
- Not using appropriate tools to assess neutralizing capacity
Recommendations/Comments to authors:
Major:
- SARS-CoV-2 Neutralization Antibody Detection Kit used by authors is just a surrogate virus neutralization test (sVNT), a serological assay to determine the presence of RBD blocking antibodies that compete for human ACE2 binding but does not assess actual virus neutralization capacity. There are several manuscripts presenting non-RBD targeting antibodies possessing neutralizing capacity (please see below for references). So, I recommend authors to include these references and discuss the same in discussion as one of the limitations of the assay or study.
- Suryadevara, Naveenchandra, Swathi Shrihari, Pavlo Gilchuk, Laura A. VanBlargan, Elad Binshtein, Seth J. Zost, Rachel S. Nargi et al. "Neutralizing and protective human monoclonal antibodies recognizing the N-terminal domain of the SARS-CoV-2 spike protein." Cell (2021).
- Chi, Xiangyang, Renhong Yan, Jun Zhang, Guanying Zhang, Yuanyuan Zhang, Meng Hao, Zhe Zhang et al. "A neutralizing human antibody binds to the N-terminal domain of the Spike protein of SARS-CoV-2." Science 369, no. 6504 (2020): 650-655.
- Usually, sVNT demonstrated a high non-neutralizing antibody detection rate. This has been evaluated in case SARS-2 sVNT kits as well and found that agreement between sVNT and PRNT-50 was moderate. Hence, conclusions about neutralization capacity are contradictory by using sVNT. I recommend authors discuss on this?
- If possible, I recommend authors to perform at least pseudo neutralization assay for few samples from each group and correlate that with sVNT assay results to check for consistency of results which will enhance the quality of the manuscript.
Author Response
Answers to the reviewer in italic.
Authors want to thank you the reviewer for the very positive feedback provided and for the suggestions for improvements as well.
Recommendations/Comments to authors:
Major:
- SARS-CoV-2 Neutralization Antibody Detection Kit used by authors is just a surrogate virus neutralization test (sVNT), a serological assay to determine the presence of RBD blocking antibodies that compete for human ACE2 binding but does not assess actual virus neutralization capacity. There are several manuscripts presenting non-RBD targeting antibodies possessing neutralizing capacity (please see below for references). So, I recommend authors to include these references and discuss the same in discussion as one of the limitations of the assay or study.
- Suryadevara, Naveenchandra, Swathi Shrihari, Pavlo Gilchuk, Laura A. VanBlargan, Elad Binshtein, Seth J. Zost, Rachel S. Nargi et al. "Neutralizing and protective human monoclonal antibodies recognizing the N-terminal domain of the SARS-CoV-2 spike protein." Cell (2021).
- Chi, Xiangyang, Renhong Yan, Jun Zhang, Guanying Zhang, Yuanyuan Zhang, Meng Hao, Zhe Zhang et al. "A neutralizing human antibody binds to the N-terminal domain of the Spike protein of SARS-CoV-2." Science 369, no. 6504 (2020): 650-655.
We appreciate this observation. In the current version we added the following text:
Our results on neutralization are built on using the RBD sequence from the authentic SARS-CoV-2 virus. “Some works demonstrated the presence of non-RBD targeting antibodies possessing neutralizing capacity (Chi et al., 2020; Suryadevara et al., 2021) that may not be detected by the surrogate assay we implemented. We may be missing the contribution of some non-RBD neutralizing antibodies, however, the RBD domain continues to be the key target for SARS-CoV-2 neutralization (Alsoussi et al., 2020; Barnes et al., 2020; He et al., 2005; Vabret et al., 2020; Zost et al., 2020). Any other additional neutralizing activity we may be missing would results in an increase of the antibodies function, reinforcing the effect we are reporting here, where function is more reliable than the presence or not of antibodies to follow up the immune response to this novel coronavirus and to the COVID-19 vaccines”.
- Usually, sVNT demonstrated a high non-neutralizing antibody detection rate. This has been evaluated in case SARS-2 sVNT kits as well and found that agreement between sVNT and PRNT-50 was moderate. Hence, conclusions about neutralization capacity are contradictory by using sVNT. I recommend authors discuss on this?
The detection of high non neutralizing antibodies by using sVNT is an interesting and essential observation.
SARS-CoV-2/COVID-19 research field has evolved at unprecedent speed. What is novel today is surpassed by some new remarkable scientific finding the following week.
This is why we decided to use an FDA-emergency use approved assay, as a tool, to test the functionality of the antibodies. Also we decided to use this assay because results obtained with it has been generally accepted by the scientific community and published in leading journal like NEJM, Lancet, Nature, and others. We quoted some of those papers in our manuscript.
Of course that “technical” challenge doesn’t change what the reviewer highlighted.
We cannot rule out that we are measuring some non-neutralizing antibodies.
Because of that, and in line with the suggestion 3 (below) we had completed an experiment, side-by-side, testing same samples by sVNT and a classical Focus Reduction Neutralization Test (FRNT) using a live virus in a BSL3 facility. See detailed answer below.
In our conditions we found a 95.12% of agreement between both techniques to detect positive samples (Cohen’s Kappa value of 0.474 for a moderate agreement).
In fact, from 41 samples tested with both assays, we found only 3 samples with FRNT titer but negative in the sVNT (FRNT+/sVNT-), suggesting the presence of some non-RBD neutralizing antibodies.
However, we did not identify any sample with the opposed profile, FRNT-/sVNT+. Because of that, we consider that, in our experimental conditions, the contribution of non-neutralizing antibodies to our results can be considered very limited.
This information is now included in the new version along with new references supporting the performance of the specific sVNT we implemented.
- If possible, I recommend authors to perform at least pseudo neutralization assay for few samples from each group and correlate that with sVNT assay results to check for consistency of results which will enhance the quality of the manuscript.
We apologize for the oversight on this particular observation.
In line with the reviewer’s suggestion, we actually had completed a comparison between the sVNT versus a classical Focus Reduction Neutralization Test (FRNT) using a live virus in a BSL3 facility. Now we realized that, by oversight, the FRNT-assay and the results descriptions are missing in the submitted version. In fact, partial results were provided in that version, as supplementary figure 3B, but were not clearly described in the text.
Only in the figure’s legend we acknowledged that: “Panel B also shown a moderate agreement between the sVNT and Focus Reduction Neutralization Test (FRNT) using the whole virus”. We also added: “A subset of 15 samples with prior known FRNT results, were used for the correlation analysis showed in panel B”.
We found a moderate agreement between the two techniques., using Cohen’s Kappa agreement following Landis and Koch scale.
On this new version the information is clearly provided in the reviewed manuscript including a separated M&M section (2.5) to describe the FRNT.
In the submitted version we showed the results of only 15 samples out of 41 tested. Initially we decided to use only those 15 because those samples were from subjects that received the vaccine later on during the study. However, considering reviewer’s observation we added the full data set from the 41 samples, pre-vaccination, that we originally used to determine the level of agreement between assays.
Reviewer 2 Report
Authors Wrote an very interesting paper. I find it well wrote, high quality and with good idea research. In my opinion the paper need only some minor revisions. Below my suggestions
- Introduction: updata data on COVID 19 burden at the day of resubmission. Furthermore, introduce better the different mechanism to development immunity response with vaccination
- Methods and results: no comment, vey interesting
- Discussion: authors wrote 4 pages of discussion. Very very well done.
- Conclusion: Give some public health proposal and perspectives that came form your excellent paper
Author Response
Answer to the reviewer in italics
Authors would like to thank you the reviewer for the strong support to our work.
We updated the data on COVID 19 burden as of today.
We also highlighted better the different mechanisms to development immunity response with vaccination